# Associations between Physical Activity and Kyphosis and Lumbar Lordosis Abnormalities, Pain, and Quality of Life in Healthy Older Adults: A Cross-Sectional Study

**DOI:** 10.3390/healthcare11192651

**Published:** 2023-09-29

**Authors:** Victoria Zaborova, Oxana Zolnikova, Natiya Dzhakhaya, Svetlana Prokhorova, Alexander Izotov, Tatyana Butkova, Vasiliy Pustovoyt, Ksenia Yurku, Dmitry Shestakov, Tatyana Zaytseva, Hassan Shafaei

**Affiliations:** 1Institute of Clinical Medicine, I.M. Sechenov First Moscow State Medical University (Sechenov University), 119991 Moscow, Russia; zaborova_v_a@staff.sechenov.ru (V.Z.); zolnikova@staff.sechenov.ru (O.Z.); dzhakhaya@staff.sechenov.ru (N.D.); prokhorova@staff.sechenov.ru (S.P.); 2Biobanking Group, Branch of Institute of Biomedical Chemistry “Scientific and Education Center”, 109028 Moscow, Russia; izotov.alexander@gmail.com (A.I.); tatyana1234501@gmail.com (T.B.); 3Laboratory of Big Data and Precision Restorative Medicine, State Research Center-Burnasyan Federal Medical Biophysical Center of Federal Medical Biological Agency, 119435 Moscow, Russia; pustovoyt126598@mail.ru (V.P.); ks_yurku@mail.ru (K.Y.); 4Moscow Clinical Scientific Center Named after A. S. Loginov, 111123 Moscow, Russia; dimitrauma@bk.ru; 5Institute of Public Health, Sechenov First Moscow State Medical University, 119991 Moscow, Russia; zaytseva_t_a@staff.sechenov.ru; 6Department of Physical Rehabilitation, Massage and Health-Improving Physical Culture Named after I. M. Sarkizov-Serazini RSUFKSMiT, 105122 Moscow, Russia

**Keywords:** elderly, physical activity, kyphosis and lumbar lordosis abnormalities, pain, quality of life

## Abstract

Significant evidence suggests that regular physical activity (PA) leads to numerous physical and psychological outcomes in the elderly. This cross-sectional study was designed to further explore this issue by investigating the associations between PA (measured by accelerometer) and kyphosis and lumbar lordosis abnormalities, pain, and quality of life (QoL) in the elderly. In this cross-sectional study, 163 older adults (73 women) over 65 years of age (mean age: 68.70 ± 3.09) from Russia acted as participants. The following instruments were used to measure research variables: ActiGraph wGT3X-BT for measuring PA, spinal-mouse for measuring kyphosis and lumbar lordosis abnormalities, and the World Health Organization Quality of Life Scale (WHOQOL-BREF) questionnaire for measuring QoL. Pain was measured using two questions. The Independent t-test and a regression analysis were used to analyze data. The results showed that our sample participated on average in 15.8 min of moderate PA (MPA) per day, which is lower than the recommended guidelines. Men were significantly more physically active than women. In addition, MPA was significantly associated with lower kyphosis and lumbar lordosis abnormalities and pain in older adults. Finally, MPA was significantly associated with higher QoL. These findings indicate that PA is a critical concern for the elderly. Accordingly, physical educators and fitness instructors should adopt appropriate strategies to promote an active lifestyle among older adults.

## 1. Introduction

Old age is a natural, biological, and general process of human life, and paying attention to the issues and needs of this stage is a social necessity. According to the reports of the World Health Organization, by 2025, the population of people aged 65 years and over will reach more than 800 million and it is worth considering that the share of developing countries in this population will be about 70% [1]. Other evidence has predicted by 2030, the ratio of the elderly population in developing countries will be nine times the current situation [2]. With the onset of old age, all kinds of physical and mental infirmities and diseases become prevalent, and this period of life is a sensitive and traumatic period due to fundamental physiological and psychological changes. The increasing sense of dependence and lack of role in life adds to the sensitivity of this era [3], despite the fact that aging is the right of all human beings and old age is not considered a disease but rather a transition from one stage to a new stage of life [4]. It is well documented that with increasing age, due to the disorders that occur in different body organs, especially due to movement limitations, a person’s dependence on others in doing daily tasks increases, and these factors can bring negative effects on the feeling of well-being and the quality of life (QoL) [5].

One of the most important aging-related changes in the human body is the breakdown and destruction of muscle mass, and a significant decrease in the volume and size of skeletal muscles, which is known as “sarcopenia” [3,4]. Sarcopenia is the loss of muscle mass, strength, and function that occurs with age. From the age of 30, muscle mass decreases between 0.1% and 0.5% per year and this process accelerates from the age of 65. Sarcopenia is associated with negative consequences including disability, weakness, illness, dependence on others, falls, fractures, and death [3,4]. In addition, among the common consequences of age-related muscle loss are musculoskeletal disorders such as kyphosis and lordosis. Kyphosis and lordosis are painful abnormalities of muscles, tendons, joints, and nerves that can affect the curve of the spine [3,4,5,6]. Symptoms of kyphosis and lordosis can be one or more complaints such as pain, tingling, stiffness or limited movement in the upper back and waist, lasting more than a week or repeating at least once a month during the past year [3,4,7]. Kyphosis and lordosis are among the multi-caused abnormalities that are caused by physical, psychological, organizational, and individual factors [8,9,10].

Concerning older adults, kyphosis and lordosis can lead to several physical and psychological health-related problems such as reduced mobility and physical independence, local pains, depression, and ultimately reduced QoL [7,10]. Also, it has been shown that older adults sometimes overestimate or underestimate their health status [11,12]. Indeed, some older adults who self-report the absence of physical defects, may have kyphosis or lordosis, both of which are significant contributors to the worsening of QoL and back pain in older adults [10]. Therefore, one should not only rely on the self-reports of older adults regarding their health status, especially kyphosis and lordosis abnormalities, rather it would be important to continuously measure the status of these abnormalities. Hence, considering the increase in the life expectancy of the elderly in recent years and, on the other hand, the increasing impact of modern life on the elderly, it is necessary to continuously measure the status of kyphosis and lordosis in older adults as well as to discover the factors that can reduce the incidence of these abnormalities and increase the QoL of this population. One of these possible factors is physical activity (PA).

It has been shown that there is a positive relationship between PA and health status by influencing morphological, muscular, motor, and metabolic aspects, and people who have regular PA maintain their physical and mental functions and lead their daily lives independently [13,14,15]. Concerning older adults, research has shown that more than half of the elderly over the age of 65 are not physically active, which can be associated with negative consequences, including disability and dependence [16,17,18,19,20,21,22]. Moreover, elderly people who have enough PA have a 20–30% lower mortality rate than those who have an inactive lifestyle [23,24]. In addition, it has been shown that regular participation in PA leads to healthy aging and increased QoL [25,26,27,28,29]. Regular participation in PA helps the elderly to maintain their independence and mobility, reduce the frequency of falls and fall injuries, improve their balance and coordination, and can improve their muscular strength and endurance [30]. Some evidence also shows that even delayed or light PA such as walking can lead to positive physical and mental health benefits [28]. Other studies have shown that older people who garden at home reported better physical and cognitive health and QoL than those who are inactive [31,32]. Regardless of the type of sport or PA, some studies have demonstrated that there are positive associations between PA and QoL in older adults [33]. Finally, it has been demonstrated that participation in PA and sports can result in lower sensitivity and occurrence of pain in older adults [34,35,36].

In addition to PA, some studies have shown that sedentary behavior (SB) is one of the factors affecting the health outcomes of older adults. SB is an integrated part of the PA pattern and is defined as an activity with very little energy expenditure (i.e., ≤1.5 METs) that is primarily undertaken while sitting or lying down [37,38]. SB is observed in a wide range of settings, including the home, traveling, and leisure time, and includes time spent sitting in front of a screen (such as watching TV), traveling in vehicles, reading, and listening to music. It has been shown that higher levels of daily SB have been associated with unfavorable health outcomes for adults, particularly older adults, including the weakening of cognitive function, a reduction in mental health and physical ability, and as a result, lower QoL [39,40,41,42,43,44]. It is therefore important to consider the association between SB and the deterioration of health in older adults.

The current study sought to expand upon the previous findings by investigating the relationships between PA and kyphosis and lumbar lordosis abnormalities, pain, and QoL in older adults. In this study, we used accelerometer devices to measure PA objectively, which are the most important tools for an accurate measurement of PA in public health research. Accelerometers are simple, light, inexpensive, and convenient tools that provide accurate data and prevent the bias that usually happens in self-report tools such as questionnaires [45]. In addition, they specify the intensity and duration of PA which can help assess health-oriented PA (for example, in the case of older adults it is 30 min of moderate PA [MPA] per day) [46]. In total, the aim of this study was to investigate the relationships between objective PA and kyphosis and lumbar lordosis abnormalities, pain, and QoL in older adults.

## 2. Methods

### 2.1. Participants

This cross-sectional study was conducted in Russia between April and June 2022. The participants of this study were 163 volunteer older adults (73 women) over 65 years of age (mean age: 68.70 ± 3.09) who were recruited using social networks. The participants had to be at least 65 years old and did not have any physical defects or diseases that interfered with their participation in the study. On the other hand, participants who did not have these conditions, as well as those with a history of severe COVID-19, were excluded from the study. Based on self-reported data, our participants had no previous physical defects or a history of severe COVID-19.

According to previous studies that used the accelerometer to measure physical activity in the elderly [18,47], 169 older adults were selected as the study sample, but 6 of them did not fully implement the accelerometer protocol and therefore were excluded from the statistical sample.

### 2.2. Data Collection

First, by conducting public calls on social networks, 215 people declared their readiness. Preliminary examinations and tests of these people were conducted through general health questionnaires, a readiness for PA protocol, and via conducting interviews and stating the merits and demerits of the research. Finally, 169 older adults were selected using the available sampling method, but 6 of them were excluded due to not completing the accelerometer protocol. Before the implementation of the accelerometer protocol, each participant completed the demographic information, and height and weight measurements were executed by the experimenter. Then, each participant fulfilled the questionnaires related to the research. Then, the information required to implement the accelerometer protocol, including how to install it on the thigh, the necessary cases to remove it, how to record the hours of sleep and wake up time, and how to clean it, if necessary, were explained to the participant. Each participant was asked to install the accelerometer on his/her thigh for one week (seven consecutive days) in order to collect information related to PA. At the end of the protocol, the device was taken from the participant and after disinfection, it was given to the next participant. During the protocol, an experimenter was in daily contact with each participant by phone so that the required information could be transferred to the participant if needed. An informed consent form and a personal information questionnaire were received from all participants, and they were promised that their information would remain confidential and only the general results of all participants would be used.

### 2.3. Measures

#### 2.3.1. Physical Activity

According to the guidelines of the World Health Organization (WHO), the elderly should undertake at least 30 min of MPA a day to enjoy the health benefits of PA [28]. In the current study, the standard of health-oriented PA was also 30 min of MPA. Therefore, in order to accurately measure the amount of MPA, we used an accelerometer (ActiGraph wGT3X-BT, Pensacola, FL, USA), which has high validity and reliability [47,48,49]. According to the standard protocol of the accelerometer, the participants of this study installed the device on their right thigh for seven days and were only allowed to remove the device when sleeping, bathing or doing other activities that damage the device. A count/min of ≥1952–5724 was used for calculating MPA [41].

#### 2.3.2. Kyphosis and Lumbar Lordosis Abnormalities

We used a spinal-mouse device (MED PRO model, Switzerland) to assess kyphosis and lumbar lordosis abnormalities. In order to measure the degree of kyphosis and lumbar lordosis, the participant was asked to spread his/her legs shoulder-width apart, with his/her knees straight and looking forward so he/she was in a completely normal position. Then the examiner stood behind the participant and first identified and marked the spinous appendage of C7 (seventh cervical vertebra) as a landmark. Then, the rolls of device were placed on the top and bottom of the C7 vertebra of the mouse, it was pulled down along the vertebral column almost to the S3 vertebra (third sacral vertebra). Then, this measurement was also performed on the bending and unfolding state of the trunk. Simultaneously, with the movement of the mouse along the vertebral column, the direction of movement, the shape of the vertebral column, the angle of each vertebra, and the size of the back curvature (from T1T2 to T12L1 level) and lumbar depression (from L1L2 to L5S1 level) were recorded on the monitor. Then, using the software of this device, the degree of kyphosis from T1 to T12 (1st to 12th dorsal vertebra) and the lumbar lordosis from L1 to L5 (1st to 5th lumbar vertebra) were extracted. This measurement was repeated three times for each participant and their average was recorded and analyzed as the degree of kyphosis and lumbar lordosis for each person.

#### 2.3.3. Pain

Pain frequency was measured by the following question: ‘Have you had pain in your back or spine in the past six months?’ [34]. Response options were ‘no’ (0), ‘seldom’ (1), ‘sometimes’ (2), ‘often’ (3), ‘daily (4)’, and ‘all the time’ (5). Pain intensity was measured by the following question: ‘How severe was the pain you had during the past six months?’ [34]. Response options were ‘none’ (0), ‘mild’ (1), ‘moderate’ (2), ‘severe (3), and ‘very severe (4)’. We calculated an alpha Cronbach coefficient to assess the reliability of this scale, where its coefficient was 0.82.

#### 2.3.4. Quality of Life

To measure the QoL of the elderly, the World Health Organization Quality of Life Scale (WHOQOL-BREF) questionnaire was used [50]. This questionnaire evaluates the four domains of physical health, psychological health, social relationships, and environmental health with 24 questions (each domain has 7, 6, 3, and 8 questions, respectively). The first 2 questions do not belong to any of the areas and evaluate the health status and QoL in general. Therefore, this questionnaire has a total of 26 questions. The score for each area is between 4 and 20, where 4 indicates the worst situation and 20 indicates the best situation. These scores can be converted into a score between 0–100. To evaluate the reliability of this questionnaire, we calculated an alpha Cronbach coefficient which was 0.88.

### 2.4. Statistical Analysis

Means and standard deviations of the MPA, kyphosis, lumbar lordosis, pain, and QoL were measured as descriptive data. Also, the data were reported as numbers (n) and percentages (%). The Kolmogorov–Smirnov test was used to check the normal distribution of quantitative data, where the data had a normal distribution (all *p* > 0.05). A regression analysis was utilized to measure the correlations between the research variables. Sex differences in PA, kyphosis and lumbar lordosis abnormalities, pain, and QoL were calculated using the Independent t-test. Finally, two groups were made based on the daily MPA. Accordingly, participants with less than 30 min of daily MPA were assigned to the “low-MPA” group and those with more than 30 min of daily MPA formed the “high-MPA” group. Independent t-tests were then used to examine whether the participants in these groups differed in terms of their kyphosis and lumbar lordosis abnormalities, pain, and QoL. The significant level was set at *p* < 0.05. The data were analyzed using SPSS version 26 (IBM, Chicago, IL, USA).

## 3. Results

The statistical sample of this study included 163 older adults over 65 years of age (73 women) whose demographic characteristics, including height, weight, BMI, financial status, education, and family status, are given in Table 1. Accordingly, it can be stated that the average age of men and women were 68.42 and 68.98 years old, respectively. In addition, the average BMI of the participants were 25.94 and 25.64 for men and women, respectively, representing a normal range (between 24 to 29). Also, most of the participants were at a medium level of financial status (73% of the total sample). In total, of the whole sample, 82% of the participants were educated to high-school level or lower and only 18% of them had a college education, representing that they were not highly educated. Finally, 62% of the participants were married.

Table 2 shows the mean and standard deviation of the PA pattern including SB, light PA, MPA, vigorous PA (VPA), and moderate-to-vigorous PA (MVPA). Sex differences in these variables are also presented. As shown, our sample spent on average 594.90 min sedentary per day, where women spent significantly more time sedentary than men (*p* < 0.001). While no significant sex differences were observed regarding light PA (*p* = 0.249), men participated in more MVA than women (19.70 vs. 11.92, respectively, *p* < 0.001)). Concerning the recommendation of the WHO for participating in at least 30 min of MPA per day for older adults over 65 years old, these results showed that our sample (both men and women) were clearly below the WHO guidelines. Similar findings were observed for VPA (*p* = 0.012) and MVPA (*p* < 0.001), where men had significantly more engagement in VPA and MVPA than women.

Table 3 shows the mean and standard deviation of kyphosis and lumbar lordosis abnormalities, pain, and QoL. Regarding kyphosis, our data showed that 22 participants (13%) were flat, 68 participants (42%) were hyper, and 73 participants (45%) were normal. Concerning lumbar lordosis, the results showed that 29 participants (18%) were flat, 72 participants (44%) were hyper, and 62 participants (38%) were normal. Regarding pain, the results of this study showed that 93 participants (57%) reported feeling pain in the past six months and 70 participants (43%) felt severe pain. Finally, the mean and standard deviation of overall QoL among our sample was 48.47 ± 16.22, representing an average level of QoL. The results of the Independent t-tests showed that there were no sex differences regarding kyphosis and lumbar lordosis abnormalities, pain and QoL (all *p* > 0.05).

The results of the regression analysis are shown in Table 4. First, it should be noted that due to the emphasis on MPA in the elderly, this index was used here as a predictor variable in the regression analysis. Accordingly, the results showed that increased MPA was significantly associated with lower kyphosis and lumbar lordosis (both *p* < 0.05). In addition, increased MPA was significantly associated with lower pain and higher QoL (both *p* < 0.05). Moreover, we included SB as a predictor variable in the analysis. The results showed that increased SB was significantly associated with higher kyphosis and lumbar lordosis (both *p* < 0.05). In addition, increased SB was significantly associated with higher pain and lower QoL (both *p* < 0.05).

Finally, as mentioned in the data analysis, we divided the participants in the “low-MPA” and “high-MPA” groups to compare their kyphosis and lumbar lordosis abnormalities, pain and QoL. Here, the results of the Independent t-tests showed that participants in the “high-MPA” group had significantly lower kyphosis (t = 3.694, *p* < 0.001), lower lumbar lordosis (t = 2.964, *p* < 0.001), lower pain (t = 3.571, *p* < 0.001), and higher QoL (t = 2.964, *p* < 0.001) than those in the “low-MPA” group (Figure 1).

## 4. Discussion

The purpose of this study was to investigate the relationships between PA and kyphosis and lumbar lordosis abnormalities, pain, and QoL in older adults. First, it should be stated that the results of this study showed that the older adults participating in the current study engaged in 15.8 min of MPA per day, which is clearly lower than the amount recommended by the WHO regarding the participation of older adults in at least 30 min of MPA per day. These results are consistent with the results of previous studies in other countries [18,19,20,21,22,51] that show the low participation of the older adults in health-oriented PA. In addition, our results showed that men engaged significantly more in MPA, VPA, and MVPA than women, which is consistent with the findings of previous studies [52,53]. Therefore, as the first suggestion resulting from the results of this study, it can be suggested that health experts should adopt appropriate solutions, strategies, and interventions to improve the PA status of older adults, especially women, so that these people enjoy the benefits of participation in regular PA.

On the other hand, some studies [22,54,55,56,57,58] have reported harmful consequences of not participating in regular PA in older adults. Some of these consequences include suffering from various diseases and several types of cancers. In line with these findings, the results of this study also showed that there is a negative and significant relationship between MPA and kyphosis and lumbar lordosis abnormalities, so that the more a person participates in MPA, the lower the rate of kyphosis and lumbar lordosis. Regular participation in PA improves and maintains muscle strength, and since kyphosis and lumbar lordosis abnormalities are mainly caused by weakness in the muscles that support the spine structure [32,34], it can be expected that participation in regular PA helps maintain muscle strength in the older adults and subsequently preserves the structure of the spine and prevents kyphosis and lumbar lordosis abnormalities. On the contrary, not participating in regular PA can lead to the destruction of the spine structure due to muscle weakness, which causes abnormalities such as kyphosis and lumbar lordosis in older adults [58].

In addition, the results showed that the older adults in this study mainly reported pain in the spine, which can be caused by the presence of kyphosis and lumbar lordosis abnormalities. In fact, some studies [34,35,36,59,60] have shown that having spinal deformities causes pain in the spine. However, according to the results of this study, more participation in MPA can cause less pain in the spine. This result is in line with the previous findings [35,36,60] and indicates that the greater the participation in MPA, the lower the presence of kyphosis and lumbar lordosis abnormalities and pain in the spine. These findings are confirmed by another finding of the current study, which showed that participants who were in the “high-MPA” group (more than 30 min of MPA per day) had significantly less kyphosis and lumbar lordosis abnormalities and pain than those who were in the “low-MPA” group (less than 30 min of MPA per day). Accordingly, it can be said that the regular participation of the elderly in health-oriented PA (at least 30 min of MPA per day) may prevent kyphosis and lumbar lordosis abnormalities and pain. Therefore, it is suggested that older adults regularly participate in health-oriented PA in order to maintain muscle strength and prevent kyphosis and lumbar lordosis abnormalities and local pain in the spine [35,60].

Moreover, regarding QoL, the results of this study showed that greater participation in MPA improved QoL of older adults. In fact, with the increase in age and subsequent loss of many motor and functional abilities in the elderly, QoL also decreases. According to the findings of the present study and previous studies, greater participation of the elderly in MPA improves QoL. These findings are confirmed by another finding of the current study, which showed that participants of the “high-MPA” group had significantly less kyphosis and lumbar lordosis abnormalities and pain than those in the “low-MPA” group. Among the possible reasons for the positive effects of PA on QoL are the existence of psychological effects, such as self-confidence, feelings of hope, and greater self-esteem, stronger social relationships, and the ability to adapt to problems, which are achieved partly due to the nature of PA and sport participation [26,27,61,62]. As well, maintaining physical functions of the muscles and internal organs can be achieved through regular participation in PA, which can positively affect QoL of older adults.

Finally, regarding SB, we found that increased SB was significantly associated with higher kyphosis, lumbar lordosis, pain, and lower QoL in older adults. These results are in accordance with those of previous studies (see [39,40,41,42,43,44] for review), indicating that SB is related to the deterioration of health in older adults. SB and wrong behavior habits during leisure time can cause adverse pressure on tissues such as joints, bones, muscles, and ligaments, and ultimately damage and create a weak posture or kyphosis and lumbar lordosis abnormalities and resulting pain, which can subsequently affect QoL of older adults. Along with the above-mentioned findings, it seems that those older adults who manifest an excess of daily SB and who perform less MPA are at greater risk of health complications. Therefore, older adults should reduce their daily SB and increase their time of being physically active in order to enjoy the benefits of an active lifestyle.

One of the strengths of the current study is the use of an accelerometer device to objectively assess PA in older adults, which prevents the bias of self-report tools. Also, by including both men and women in the sample, it was possible to investigate sex differences. However, with 163 participants, the sample size should be considered critically; hence, further studies with larger sample sizes are needed to discover the associations between PA with health-related factors in older adults. Moreover, our sample is not representative of the older Russian population. In addition, the sample participants of this study self-reported that they had no physical defects. However, the measurements showed that a large percentage had kyphosis and lordosis. Therefore, in future studies, it is necessary not only to rely on the self-report of the participants, but to carry out more detailed preliminary investigations on the participants. Finally, due to the cross-sectional nature of the current study, it was not possible to examine the causes of the impact of PA on kyphosis and lumbar lordosis abnormalities and QoL of the older adults. Thus, future studies should focus on finding effective causal factors for health in older adults using interventional approaches.

## 5. Conclusions

Due to the lack of studies on the impact of regular PA in older adults, the purpose of this study was to investigate the relationships between PA and kyphosis and lumbar lordosis abnormalities, pain, and QoL in the elderly. The findings showed that the older adults in the current study had low health-oriented PA (less than 30 min of MPA per day), which requires a special focus on interventions and strategies to improve the level of PA of the older adults. Also, MPA had significant relationships with lower kyphosis and lumbar lordosis abnormalities, pain, and higher QoL in older adults. These findings, in general, show that PA is a critical factor for the older adults. This study also has some practical implications. For example, it is essential for physical educators and sports coaches to design and implement interventions to improve the PA status of older adults, especially women. In addition, considering the impact of PA on kyphosis and lumbar lordosis abnormalities, it is necessary for occupational therapists to use exercise interventions to prevent and improve the musculoskeletal status of older adults. Finally, it can be stated that this study was conducted in Russia. However, considering the cultural differences, the results of this study can be used in other countries as well.

## Figures and Tables

**Figure 1 healthcare-11-02651-f001:**
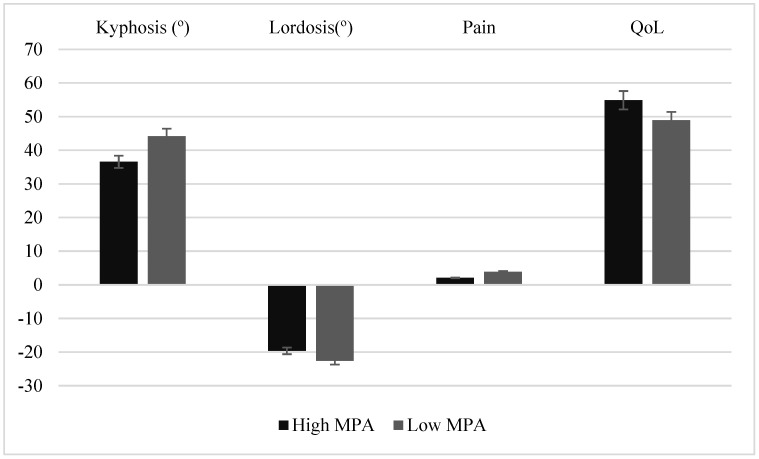
Mean and standard deviation of kyphosis, lordosis, pain, and QoL across “high-MPA” and “low-MPA” groups.

**Table 1 healthcare-11-02651-t001:** Demographic characteristics of the study’s sample.

Variables	Men (*n* = 90)	Women (*n* = 73)	Total (*n* = 163)
Age (year)	68.42 ± 2.82	68.98 ± 3.09	68.70 ± 3.09
BMI	25.94 ± 2.05	25.64 ± 2.48	25.79 ± 2.26
Financial status			
Low	16 (18%)	12 (16%)	28 (17%)
Medium	64 (71%)	55 (75%)	119 (73%)
High	10 (11%)	6 (9%)	16 (10%)
Education			
High-school and less	73 (81%)	60 (82%)	133 (82%)
College	17 (19%)	13 (18%)	30 (18%)
Family status			
Married	56 (62%)	45 (62%)	101 (62%)
Divorced	34 (38%)	28 (38%)	62 (38%)

**Table 2 healthcare-11-02651-t002:** Mean and standard deviation of PA pattern across males and females.

Variable	Men (*n* = 90)	Women (*n* = 73)	Total (*n* = 163)	Sex Differences
Sedentary time (minutes/day)	539.16 ± 195.11	650.64 ± 232.84	594.90 ± 213.74	t = −9.684 *p* < 0.001 **
Light PA (minutes/day)	122.84 ± 21.08	128.76 ± 19.50	125.80 ± 19.93	t = 0.265 *p* = 0.649
MPA (minutes/day)	19.70 ± 8.27	11.92 ± 5.94	15.81 ± 7.03	t = 3.422 *p* < 0.001 **
VPA (minutes/day)	3.97 ± 2.54	2.19 ± 2.49	3.08 ± 2.50	t = −2.795 *p* = 0.012 *
MVPA (minutes/day)	23.67 ± 6.29	14.11 ± 3.91	18.89 ± 5.05	t = 6.749 *p* < 0.001 **

* Significant at *p* < 0.05; ** significant at *p* < 0.001.

**Table 3 healthcare-11-02651-t003:** Mean and standard deviation of research variables across males and females.

Variable	Men (*n* = 90)	Women (*n* = 73)	Total (*n* = 163)	Sex Differences
Kyphosis (°)	41.49 ± 3.64	39.69 ± 3.64	40.59 ± 3.64	t = −0.948 *p* = 0.204
Lumbar lordosis (°)	−27.69 ± 2.34	−26.57 ± 2.07	−27.13 ± 2.19	t = 0.394 *p* = 0.582
Pain (intensity)	3.08 ± 1.08	2.82 ± 1.09	2.95 ± 1.22	t = 0.509 *p* = 0.327
QoL (total score)	49.67 ± 18.27	47.28 ± 15.17	48.47 ± 16.22	t = −0.641 *p* = 0.297

**Table 4 healthcare-11-02651-t004:** Results of correlations between MPA and SB with kyphosis, lumbar lordosis, pain, and QoL.

	Variable	Adjusted OR	95% CI	*p*
MPA	kyphosis	1.82	1.30–1.93	<0.001
	lumbar lordosis	1.65	1.60–2.28	<0.001
	pain	1.72	1.33–2.08	<0.001
	QoL	1.80	1.39–2.40	<0.001
SB	kyphosis	1.64	0.98–2.60	<0.001
	lordosis	1.55	0.69–2.33	<0.001
	pain	3.25	0.38–5.71	<0.001
	QoL	1.98	0.96–2.67	<0.001

## Data Availability

The data presented in this study are not publicly available due to ethical reasons.

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
