# Peer review of "Associations between Physical Activity and Kyphosis and Lumbar Lordosis Abnormalities, Pain, and Quality of Life in Healthy Older Adults: A Cross-Sectional Study"

_healthcare, 2023, doi:10.3390/healthcare11192651_

Round 1
Reviewer 1 Report
My comments are listed below.
Abstract
- Line 24: Please add the mean age of the participants.
- Line 27: Please add the whole name of the questionnaire for the first time.
- May you please add most important practical implication of the results of this study?
Introduction
- Line 56: Since the participants of this research are elderly people over 65 years old, an important phenomenon in this age group is sarcopenia. Please explain this phenomenon further.
- Since research on factors related to the elderly has grown significantly in recent years, it is necessary to use recent references (mainly related to the last 10 years) in the introduction.
Method
- Is there any info related to the health status of the participants? Please explain.
- Please add a reference (citation) for the measurement of pain.
Results
- Results are well structured and formulated.
Discussion & Conclusion
- The results are well discussed and concluded, however, please extend strengths and limitations of the study.
- Please add some practical implications for the findings of this paper.
Best regards,
Reviewer 2 Report
Congratulations to the authors for the topic under study. However, this study presents some issues that may need to be addressed. In the comments below, some considerations are set out.
Although the title informs us of all the variables under study and that this is an observational study, we must consider that there are a wide variety of research methods. As the title is of vital importance when selecting and reading research reports, it is recommended to identify the report in terms of its research design.
The "Introduction" section contextualisation of the object of study. However, it is suggested that the authors could also integrate the concept of sedentary behaviour. Currently, it is necessary to take into account a balanced approach to physical activity, sedentary behaviour, and sleep (https://doi.org/10.1139/apnm-2020-0467). We also suggest updating the operational definition of sedentary behaviour (https://doi.org/10.1186/s12966-017-0525-8). In reality, all participants manifest sedentary behaviour on a daily basis, even those who can be considered physically active. Therefore, sedentary behaviour is a different concept from physical inactivity.
We suggest presenting the psychometric validity of the questionnaire applied to "Pain".
The participants were divided into two groups based on the number of minutes of daily PA. However, it's not clear which PA intensities were considered in this division of groups. Was Light PA, MPA, VPA or MVPA considered?
The "Discussion" section could present a more comprehensive and theoretical development of how excessive daily sedentary behaviour may or may not influence the results. Higher levels of daily SB have been associated with significant health outcomes for adults, particularly older adults. Furthermore, the health consequences of excessive daily sedentary behaviour seem to be more evident in physically inactive individuals. It is therefore important to consider the relationship between sedentary behaviour and moderate to vigorous physical activity in relation to the deterioration of health in older individuals. Therefore, it seems that those individuals who manifest an excess of daily sedentary behaviour and who perform less moderate to vigorous intensity physical activity are at greater risk of health complications.
Round 2
Reviewer 2 Report
The authors have taken most of the recommendations for changes into consideration. However, it is also suggested that the manuscript could be improved by including some reference works on the study of sedentary behaviour (e.g. https://doi.org/10.1146/annurev-publhealth-040119-094201).